# Diselenide crosslinks for enhanced and simplified oxidative protein folding

Reem Mousa[1], Taghreed Hidmi [1], Sergei Pomyalov [1], Shifra Lansky[1], Lareen Khouri[1], Deborah E. Shalev[2], Gil Shoham[1✉] & Norman Metanis[1✉]

The in vitro oxidative folding of proteins has been studied for over sixty years, providing critical insight into protein folding mechanisms. Hirudin, the most potent natural inhibitor of thrombin, is a 65-residue protein with three disulfide bonds, and is viewed as a folding model for a wide range of disulfide-rich proteins. Hirudin's folding pathway is notorious for its highly heterogeneous intermediates and scrambled isomers, limiting its folding rate and yield in vitro. Aiming to overcome these limitations, we undertake systematic investigation of diselenide bridges at native and non-native positions and investigate their effect on hirudin's folding, structure and activity. Our studies demonstrate that, regardless of the specific positions of these substitutions, the diselenide crosslinks enhanced the folding rate and yield of the corresponding hirudin analogues, while reducing the complexity and heterogeneity of the process. Moreover, crystal structure analysis confirms that the diselenide substitutions maintained the overall three-dimensional structure of the protein and left its function virtually unchanged. The choice of hirudin as a study model has implications beyond its specific folding mechanism, demonstrating the high potential of diselenide substitutions in the design, preparation and characterization of disulfide-rich proteins.

[1] Institute of Chemistry, The Hebrew University of Jerusalem, Jerusalem, Israel. [2] Department of Pharmaceutical Engineering, Azrieli College of Engineering Jerusalem and Wolfson Centre for Applied Structural Biology, The Hebrew University of Jerusalem, Jerusalem, Israel. ✉email: Gil2@mail.huji.ac.il; Metanis@mail.huji.ac.il

The in vitro oxidative folding of many proteins is rather a sluggish process, often making it a major bottleneck in the preparation of therapeutic, disulfide-rich proteins[1,2]. During this process, a reduced/unfolded polypeptide undergoes both conformational folding and intermediate disulfide bond formation, eventually reaching the native state (N) of the protein with its final disulfide crosslinks[3–5]. On the other hand, the slow kinetics of disulfide bond formation and shuffling enable chemical trapping, isolation and further characterization of the folding intermediates[3,6], allowing an important in-depth elucidation of the in vitro folding process, which may also shed light on the protein folding in vivo[7]. Extensive studies of such processes have shown divergent in vitro oxidative folding pathways that differ mainly in the heterogeneity of the intermediates[3–5]. At the end of this wide spectrum of processes, there are two extreme mechanism models, represented by bovine pancreatic trypsin inhibitor (BPTI) and hirudin[3–5]. BPTI folding, which represents one extreme of in vitro folding, is characterized by the predominance of a finite number of distinct intermediates. These intermediates have native-like structures, which assume one, and then two, native disulfide bonds, before reaching the final conformational state of the protein with its three native disulfide bonds[8–10]. The other extreme model is hirudin, a small protein isolated from the leech *Hirudo medicinalis*[11], which is known to be the most potent natural inhibitor of thrombin[12,13]. Hirudin contains two major parts, a compact N-terminal domain, held by three disulfide crosslinks, and an extended, practically unstructured, C-terminal peptide (Fig. 1a)[14]. Unlike BPTI, the folding pathway of hirudin involves numerous, highly heterogeneous disulfide-bonded intermediates, generally known as a "trial and error" mechanism, where 75 different possibilities of disulfide-

containing crosslinks could form, including 15 different, fully oxidized isomers (3-SS)[4,15,16]. Reduced hirudin folds into its final native state in two major stages. First, the protein undergoes a non-specific packing stage, in which the protein pairs random disulfides to form heterogeneous intermediate ensembles, involving a single disulfide (1-SS), two disulfide (2-SS) and three disulfide (3-SS) scrambled species (Fig. 1b). The second, and rate-determining stage, is the disulfide reshuffling of the heterogeneous scrambled population, leading to the formation of the final structure with its three native disulfide bonds, 6–14, 16–28 and 22–39 (C1–C2, C3–C5, C4–C6) (Fig. 1a)[4,15,16].

Scrambled isomer formation and the accumulation of trapped intermediates are known to decrease the efficiency of in vitro oxidative folding of many proteins, thereby limiting their folding rate and yield[17]. One of the potential approaches to overcome such limitations involves the use of selenium-containing molecules, which can act as inter/intramolecular catalysts in disulfide exchange reactions[18–25]. A similar approach involves cysteine to selenocysteine (Sec, U) substitutions, which has been used to study the oxidative folding of several proteins[22,26–32]. In fact, this isosteric replacement strategy was applied to native and non-native disulfides in hirudin's opposite model, BPTI[32–35]. In that study, the seleno-BPTI (Se-BPTI) analogues were found to fold correctly into the native state, some of which were demonstrated to bypass the common trapped intermediates of the folding pathway[34]. These improved folding results of the Se-BPTI case study motivated us to investigate the effect of diselenide substitution on the folding pathway of hirudin, which has a drastically different folding pattern. Given the more chaotic folding pathway of hirudin, we speculated that the introduction of diselenide crosslinks within the hirudin protein would decrease the distribution of folding intermediates, and thereby improve the overall folding rate. That is, by minimizing the random disulfide parings inherent to the native folding pathway, it was expected that the productive folding routes would be enhanced over the unproductive ones, leading to a more efficient folding process.

In the present work, following the above rationale, we prepared and characterized four seleno-hirudin (Se-Hir) analogues in order to study the effect of Sec substitutions on the folding, structure and biological activity of the protein. Three of the analogues contained diselenides at the native crosslinks, 6–14, 16–28 and 22–39, while the fourth analogue introduced a diselenide bond at a non-native crosslink, 6–16, due to the suggested role of this non-native disulfide bond in the early stage of hirudin folding[17,36]. Overall, our results demonstrate that disulfide-to-diselenide substitutions of the native disulfide crosslinks improve the folding efficiency to the native state, significantly minimizing the formation of non-productive intermediates. Remarkably, this was also the case with the non-native diselenide-containing analogue 6–16. Two of the Se-Hir analogues studied here indicated a simpler folding pathway, where the previously common 1-SS intermediate ensembles were largely not observed (see below). Interestingly, the overall 3D structure of all the Se-Hir analogues studied here remained almost unchanged, as confirmed unequivocally by crystal structure studies of the corresponding Se-Hir-thrombin complexes. While two of the Se-Hir analogues showed thrombin inhibitory activity very similar to WT-Hir, two other analogues exhibited a slightly lower inhibitory activity, probably as a result of small local conformational changes around the substituted selenium atoms, mainly at positions 6, 14 and/or 16.

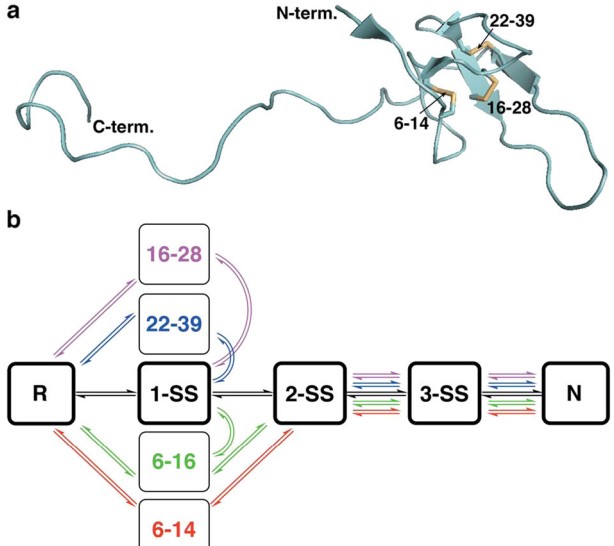

**Fig. 1 The 3D structure and the 1proposed oxidative folding mechanism of hirudin. a** 3D structure of WT-hirudin (WT-Hir; PDB entry: 1HRT) based on crystal structure of its complex with bovine thrombin (not shown here). The three native disulfide bonds 6–14, 16–28 and 22–39 are shown as sticks (yellow). **b** The proposed oxidative folding mechanism of WT-Hir (in black) and its seleno-analogues (coloured).[12,13] R and N are reduced/unfolded and native hirudin, respectively; 1-SS, 2-SS and 3-SS are ensembles of molecules with the corresponding number of disulfide bonds. The four synthetic Se-Hir analogues (coloured) contain a diselenide bond at the specified positions. All Se-Hir analogues folded faster than WT-Hir, and with fewer numbers of intermediates. Although all analogues formed the 2-SS and 3-SS ensembles, the 1-SS populations were mostly absent in Hir (C6U/C14U) and Hir(C6U/C16U) analogues.

## Results

**Chemical synthesis of WT-Hir and its seleno-analogues.** For this study, we first chemically synthesized wild-type hirudin (WT-Hir), followed by the four Se-Hir analogues, as described

**Fig. 2 Chemical synthesis of WT-Hir and its seleno-analogues.** The WT-Hir, Hir(C6U/C14U), Hir(C16U/C28U), Hir(C22U/C39U) and Hir(C6U/C16U) have been produced by native chemical ligation (NCL) reactions. The amino acid sequence of WT-Hir is also shown. The ligation site is indicated in bold/ underlined letters, while the Cys residues are marked in bold.

above. All five proteins were chemically prepared from two segments, using a single native chemical ligation (NCL)[37,38] reaction at the Gln38-Cys39 junction (Fig. 2), which is roughly located at the middle of the protein sequence[39]. The corresponding C-terminal peptides Hir(39-65), containing Cys/Sec39, were synthesized by Fmoc-SPPS[40,41], purified and characterized by HPLC and ESI-MS (Figs. S1 and S2). The Hir(1-38)-COSR N-terminal peptides were similarly prepared by Fmoc-SPPS, bearing a C-terminal thioester surrogate (either hydrazide or *N*-acylurea derivative (Nbz))[42,43] and Sec at different specific positions (6, 14, 16, 22 and/or 28) as needed (for details see supplementary methods 4.1, and Figs. S3–S7). Cys/Sec-NCL[44–47] was then performed for the preparation of all the analogues studied (Figs. S8–S12), providing milligram quantities of the homogenous proteins. For the Se-Hir analogues this resulted in proteins with a single diselenide crosslink and four reduced thiols, while the WT-Hir protein was isolated in the fully reduced form, as confirmed by MS analysis (see the supplementary methods 4.1 for further information).

**Oxidative folding studies**. Once the proteins were synthesized and purified, they were subjected to oxidative folding under anaerobic conditions at pH 8.7, using oxidized glutathione (GSSG) as the "folding catalyst" (see details in the supplementary

methods 4.3). All folding experiments were performed according to the previous study by Chang et al. to allow for a direct comparison[16]. For WT-Hir, under these conditions, the common three ensemble populations, namely 1-SS, 2-SS and 3-SS, could be clearly observed during the folding process. These species overlapped extensively and hence appeared in the HPLC output as broad peaks (Fig. 3a), in good agreement with previous reports[16,48]. Within 0.5 h, the 3-SS species predominated and remained stable even after 1.5 h[17,49]. The final stage of reshuffling ended only after 7 h, yielding about 88% of the native form of WT-Hir (Table 1, Figs. 3a and S20).

For the Se-Hir analogue Hir(C16U/C28U), a significantly faster folding process was observed under the same conditions, as compared to WT-Hir. More than half of the protein reached the native state within only 0.5 h, and the entire folding process was completed within 1.5 h, yielding about 90% of the native form of the protein (Table 1, Figs. 3b and S21c). The intermediates of this process appeared as fewer and more distinct peaks, containing mainly 2- and 3-crosslinks. This pattern indicates a significant decrease in the heterogeneous populations (1-SS, 2-SS and 3-SS), where scrambled intermediates predominate and hence rearrange more rapidly to the final native state. Thus, Sec substitution at positions 16 and 28 lead to a fivefold increase in the folding rate relative to WT-Hir, with comparable overall yield, and is in good

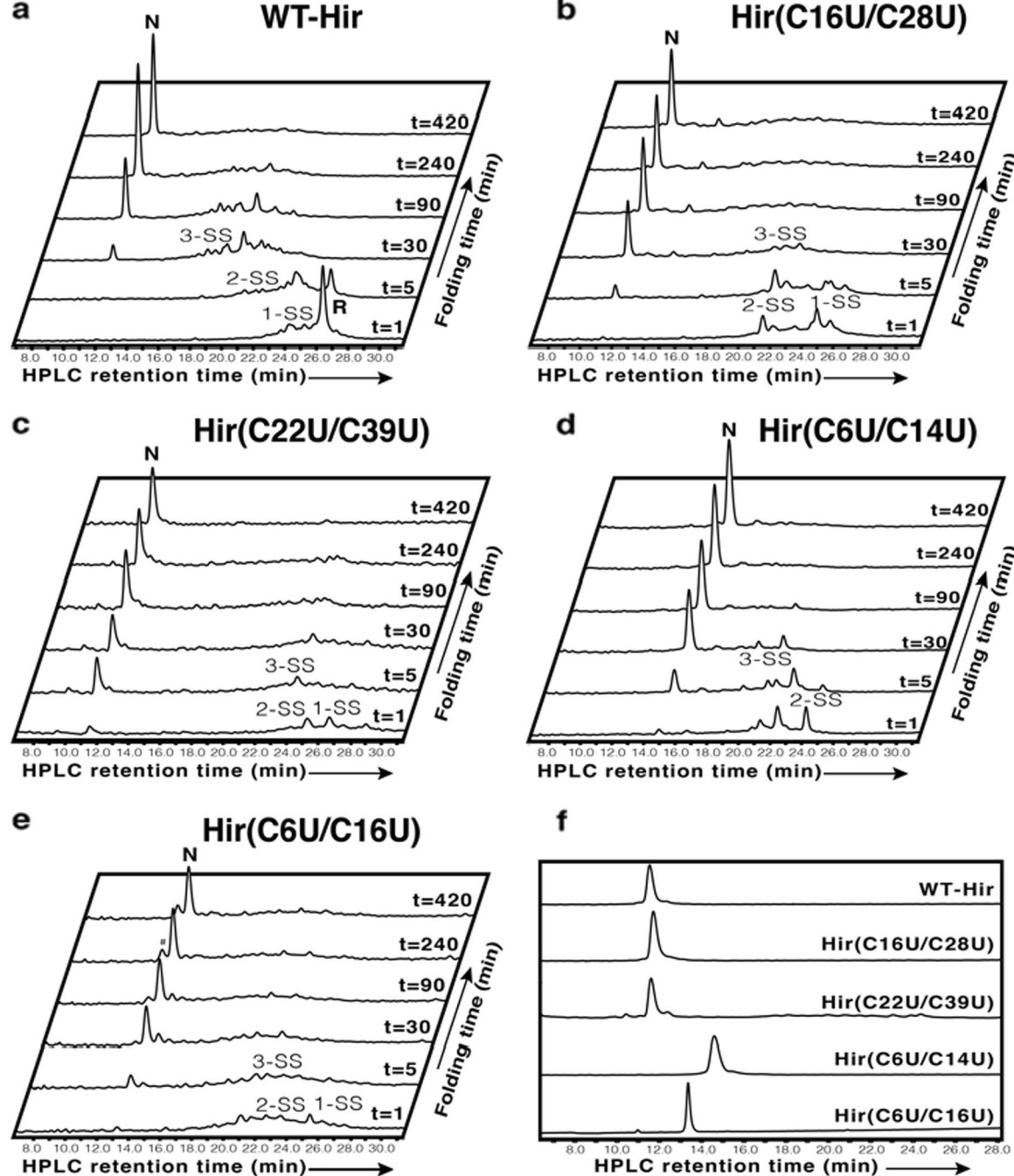

**Fig. 3 Oxidative folding analysis of the synthetic proteins.** The anaerobic oxidative folding at pH 8.7 in the presence of 150 μM GSSG and 30 μM of **a** WT-Hir, **b** Hir(C16U/C28U), **c** Hir(C22U/C39U), **d** Hir(C6U/C14U) and **e** Hir(C6U/C16U). # denotes an impurity with unknown mass. 1-SS, 2-SS and 3-SS represent the number of disulfide (or diselenide) crosslinks in the intermediates. **f** Retention times of isolated native states of WT-Hir and the seleno-variants.

correlation with the initial predictions made above. The advantage of using Sec in oxidative folding is even more significant under aerobic conditions and in the absence of additional oxidants[31,34]. Air oxidation of Hir(C16U/C28U) resulted in 80% overall yield of folded protein after 5 h, while under the same conditions, the folding of WT-Hir resulted in only 27% overall yield of the folded protein after 22 h (Table 1, Figs. 4a and S13). These results also emphasize the importance of GSSG as a component in oxidative protein folding, as it appears to be critical not only as an oxidant but also as a catalyst for the disulfide (and diselenide) reshuffling, through the release of

GSH[3–5]. Additionally, it is worth noting that although Sec substitution steers the folding process to form higher ratios of productive intermediates, a small number of trapped unproductive intermediates could be observed alongside the main process, which appear to be stable even after 7 h. This indicates that scrambled isomers of this Se-Hir analogue react differently towards reshuffling[17], and a small fraction becomes even more stable as a result of the Cys to Sec substitutions.

The Se-Hir analogue Hir(C22U/C39U) also showed a significant improvement in folding efficiency under anaerobic conditions. About 66% of the protein reached the native state

**Table 1 Characterization of WT-Hir and the Se-Hir analogues.**

| Protein | Folding time (min) | Yield% by HPLC integration (isolated) | HPLC r.t. of N[a] (min) | $K_I$ (pM) [b] |
|---|---|---|---|---|
| WT-Hir | 420 | 88% (61 ± 1%) | 11.2 | 10.9 ± 4.9 |
| WT-Hir (w/o GSSG, aerobic) | 1320 | 27% (n.d.) | 11.2 | n.d. |
| Hir(C16U/C28) | 240 | 90% (56 ± 1%) | 11.3 | 10.0 ± 3.7 |
| Hir(C16U/C28) (w/o GSSG, aerobic) | 300 | 80% (n.d.) | 11.3 | n.d. |
| Hir(C22U/C39U) | 90 | 94% (61 ± 1%) | 11.3 | 12.5 ± 2.9 |
| Hir(C6U/C14U) | 90 | 95% (66 ± 1%) | 14.8 | 192.4 ± 21.9 |
| Hir(C6U/C16U) | 240 | 90% (46 ± 1%) | 13.2 | 104.9 ± 15.0 |

The oxidative folding studies were performed under anaerobic conditions (expect if stated otherwise) in Tris-HCl buffer at pH 8.7 with 30 µM protein with 150 µM GSSG. The WT-Hir and Hir(C16U/C28U) were also folded under aerobic conditions in the absence of GSSG [a] the retention time (r.t.) of the native state (N) is shown in Fig. 3f. [b] The inhibitory activity of the different hirudin analogues with thrombin was assayed in buffer at 37 °C and using *N*-(*p*-Tosyl)-Gly-Pro-Arg-*p*-nitroanilide as a substrate, where the initial rate of *p*-nitroaniline formation was followed at 405 nm by UV (complete details in the Supplementary Methods 4.6).

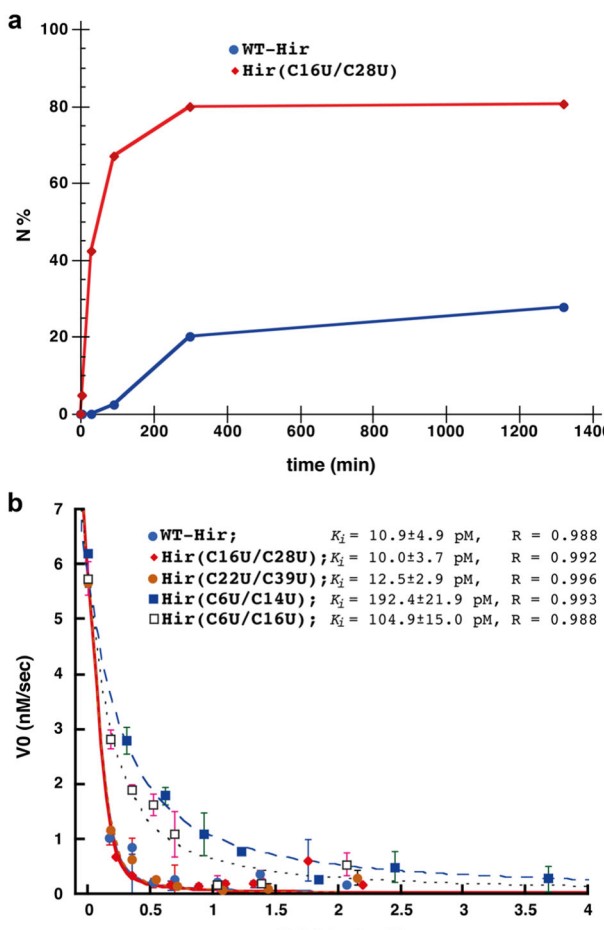

**Fig. 4 Comparison between the synthetic proteins; aerobic folding and inhibition of thrombin. a** Kinetic traces of the aerobic folding in the absence of GSSG and with 30 µM WT-Hir (blue circles) or Hir(C16U/C28U) (red diamonds) at pH 8.7. The lines connecting the data points are shown only for illustrative purposes and do not represent a data fit. The data points are based on two repetitive experiments and the error was within 5%. **b** Inhibition activity of WT-Hir and Se-Hir analogues. The inhibition of bovine thrombin by WT-Hir (blue circles), Hir(C16U/C28U) (red diamonds), Hir(C22U/C39U) (brown circles), Hir(C6U/C14U) (blue filled squares) and Hir(C6U/C16U) (open squares). The data were fitted as described in the Supplementary Methods 4.6 to give the apparent $K_I$ values indicated.

within 5 min, and the entire folding process was completed within 1.5 h, yielding about 94% of the native form of the protein (Table 1). Interestingly, a major HPLC peak was detected at 23 min, which contained mainly the 3-SS intermediate species (Figs. 3c and S21d).

Although the folding yield of the Hir(C6U/C14U) proved to be similar to that of other proteins (Table 1, Figs. 3d and S21b), yet, this analogue formed significantly fewer intermediate peaks during the folding process (Fig. 3d). These results are in good correlation with previous kinetic and computational studies, which indicated that not all the disulfide crosslinks have a similar role during the folding process of hirudin. The 6–14 crosslink has been suggested to play a critical role, since it was found to be highly populated among the 1-SS, 2-SS and 3-SS species in WT-Hir[17,36]. Unlike WT-Hir, Hir(C16U/C28U) and Hir(C22U/C39U), no 1-SS species were detected during the folding process of Hir(C6U/C14U) (Fig. 3d). This can be explained by a preexisting 6–14 diselenide crosslink[36], which, once present, is likely to more easily govern the formation of the remaining disulfide bonds. Thus, these current results further confirm the critical role of the 6–14 disulfide bond in the hirudin folding pathway (Fig. 1b).

**Biological activity**. Despite its favourable folding rate and yield, we found that the Hir(C6U/C14U) analogue displayed a lower inhibitory activity toward its natural target thrombin. The inhibitory constant $K_I$ of this analogue for thrombin was about 20-fold higher than either WT-Hir[13,50] or the other seleno-analogues, Hir(C16U/C28U) and Hir(C22U/C39U) (Fig. 4b, Table 1). Additionally, a distinct difference was observed in the HPLC retention time of the folded form of the Hir(C6U/C14U) analogue, as compared to the other seleno-variants (Fig. 3f). Although similar variations in chromatography have been reported by Alewood et al. of the seleno-analogues of conotoxins[29], the differences we observed in the retention time, combined with the lower activity against thrombin raised our concern at to whether the Hir(C6U/C14U) analogue had correctly folded into the native state.

**Structural studies**. Thus motivated to explore the folded states, we examined the detailed 3D structures of these new synthetic proteins listed above. Since free hirudin proteins proved to be quite challenging to crystallize and analyze, we undertook a comprehensive structural analysis of the relevant hirudin analogues using the crystal structures of their bound complexes with thrombin[14,51]. The three Se-Hir analogues described above were incubated with bovine thrombin to form the corresponding complexes, namely Hir(C16U/C28U)-thrombin (Complex-1),

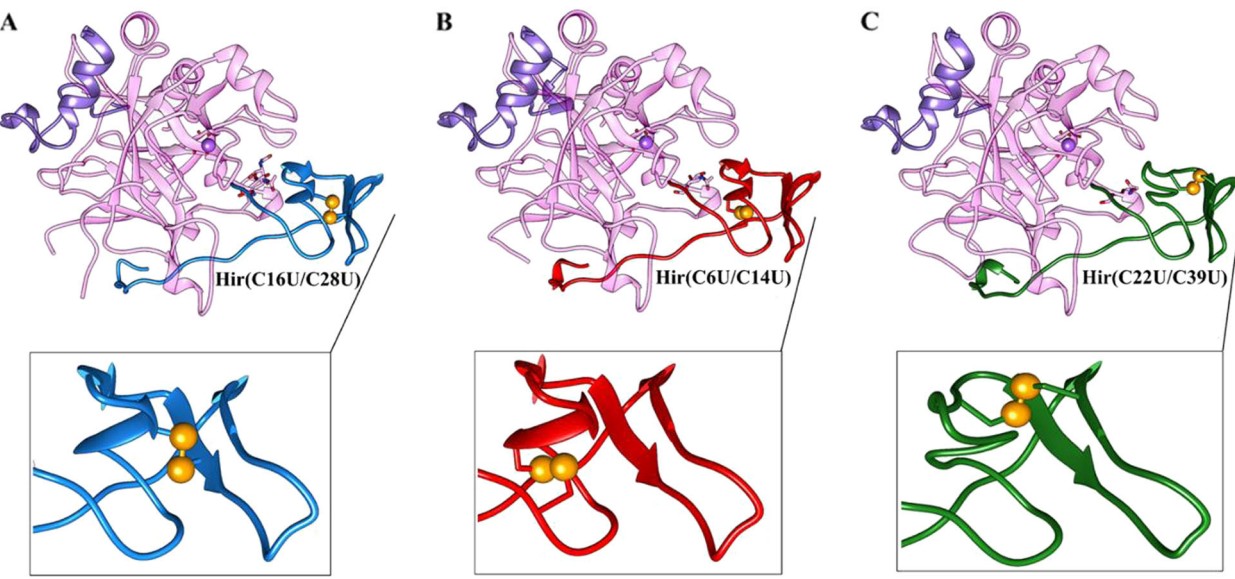

**Fig. 5 The overall structures of the three Se-Hir-thrombin complexes. A** Complex-1; **B** Complex-2; **C** Complex-3. All complexes are shown in the same orientation, where the thrombin active site is at the bottom-right, opening to the right. Colour codes: thrombin heavy chain (Orchid); thrombin light chain (purple); Hir(C16U/C28U) (blue); Hir(C6U/C14U) (red); Hir(C22U/C39U) (green). Gold spheres represent the Se atoms and the Se–Se bonds. A zoom onto the Se–Se bond is shown at the bottom of each structure, in a slightly different orientation where the diselenide bond is best viewed. The PDB IDs for Complexes 1–3 are 7A0D, 7A0E and 7A0F, respectively.

Hir(C6U/C14U)-thrombin (Complex-2) and Hir(C22U/C39U)-thrombin (Complex-3). These complexes were crystallized and their detailed 3D structures were determined and analyzed (see details in the supplementary methods 4.4). In general, all three structures were found to be quite similar to each other (Fig. 5 and Table S1), and to the reported 3D structure of the WT-Hir with bovine thrombin (PDB ID: 1HRT; Figs. 1a and S14)[52]. A global structural comparison of the bound Se-Hir in Complexes 1–3 to the bound WT-Hir in the reference structure (PDB ID: 1HRT) gives relatively small (overall) RMSD values of 1.620, 1.476 and 1.472 Å, respectively. These values, as well as a visual superposition of each of these pairs (e.g. Fig. S14) indeed demonstrate that no major global structural differences could be observed between the four structures compared here. This confirms unequivocally that, despite the replacement of a disulfide bond with a diselenide bond at the three different positions, all three Se-Hir analogues achieved eventually the native folded state, adopting practically the same overall 3D structure.

A closer look at the Se-Hir analogues within these complexes, and in comparison to WT-Hir, confirmed that most of the intramolecular hydrogen bonds have been closely maintained, especially the tight cluster of H-bonds located generally around the 6–14 disulfide (or diselenide) crosslink (Figs. S16, S17). A superposition of the Se-Hir structures in Complex-1 and Complex-2 (Table S3 in the supplementary discussion 4.4.4.2) demonstrates that this central H-bonding cluster remains practically unchanged and emphasizes its central role in the stabilization of the core structure of hirudin (Fig. S17). Especially critical within this cluster are the interactions made by Lys47 of hirudin, which plays a key role in positioning the hirudin N-terminal and C-terminal peptide segments in their specific (effective) orientations (Fig. S17). These results reinforce previous studies, which showed that the hydrogen bonds formed by the sidechain amino group of Lys47 with the backbone carbonyl of Asp5 and the hydroxyl sidechain of Thr4 help to place the otherwise rather flexible N-terminal segment of hirudin in the active-site cleft of thrombin[51]. Interestingly, only relatively small local conformational differences are apparent around the three

original disulfide bonds that have been substituted with diselenides (Fig. S15), perhaps as a result of changes in the corresponding bond lengths, dihedral angles and atom sizes (Table S2). These local changes appear to be critical in the case of the Hir(C6U/C14U) analogue, probably due to the pivotal location of the 6–14 crosslink within the central H-bonding cluster (Fig. S17; Table S4). This was further supported by 2D $^1$H-NMR COSY, in which some deviations in chemical shift by NMR were observed in the region around Thr4, Gln11, Leu13, Leu15 and Cys16 (Fig. S19).

It should be noted, however, that the similarities observed in the current structures are based on the hirudin inhibitors when they are tightly bound to their target protease, thrombin (e.g. Fig. S16 and Table S5). This is an important point, since the exact structure of the specific hirudin analogue examined here may, in principle, be different when it is unbound, since such protein-inhibitor binding may force the bound hirudin into the specific conformation that is energetically preferred by the tight hirudin-thrombin interactions. That being said, the 2D $^1$H-NMR COSY analysis confirmed that the solution structures of the free Hir (C6U/C14U) and Hir(C6U/C16U) analogues are practically identical to the corresponding free WT-Hir under the same conditions, generally addressing this important potential concern.

Along the rationale outlined above, it was interesting to investigate how a non-native diselenide crosslink influenced the hirudin folding process. Previous studies on apamin and BPTI, where non-native diselenide substitutions were introduced, showed somewhat contradictory results. While a non-native diselenide substitution impeded the formation of the native species in the short polypeptide apamin[27], a BPTI analogue folded neatly into its native state with one disulfide and two selenylsulfide crosslinks[34]. In the case of hirudin, we selected the non-native crosslink 6–16 due to its predominance in 1-SS and 2-SS ensembles (see above)[36]. Our experiment demonstrated that the Hir(C6U/C16U) analogue folded in a similar pathway to WT-Hir, forming a heterogeneous mixture of species, mainly the 2-SS and 3-SS intermediates, but rapidly converted into the native state, with one disulfide and two selenylsulfide bonds, with a yield

similar to WT-Hir (Table 1, Figs. 3e and S21e). This analogue showed a 10-fold reduction in its inhibitory activity toward thrombin (Table 1, Fig. 4b), similar to Hir(C6U/C14U), supporting our previous interpretation that a fine-tuned structural perturbation around Cys/Sec at 6, 14 and/or 16 are likely to be responsible for the slight decreased activity observed for these two analogues (see above). As with the case of the Hir(C6U/C14U) analogue, the native state of Hir(C6U/C16U) eluted at later retention time compared to WT-Hir, Hir(C16U/C28U) and Hir(C22U/C39U) (Fig. 3f, Table 1). This behaviour is most probably caused by their different hydrophobicity, stemming from the specific position of the Sec residues, compared to the other analogues, at least in the fully oxidized but unfolded state, as the condition used for the HPLC characterization were very acidic (solvents pH ~2)[29].

## Discussion

In summary, we studied the effect of diselenide substitutions on the folding of the model protein hirudin. We show that diselenide crosslink generally reduces the complexity and heterogeneity of the folding pathway of hirudin and thereby enhances the folding rate and its efficiency. Interestingly, in the case of the native 6–14 and non-native 6–16 diselenide substitutions, the 1-SS populations were mostly absent, showing that the early stages of hirudin folding could be bypassed entirely[17,36] and still resulting in the native state. However, these two diselenide substitutions slightly reduced the inhibitory activity of the corresponding hirudin analogues, likely due to subtle structural perturbations surrounding the Cys/Sec6,14 and/or 16 residues (Fig. S15). The substitution of the other two native crosslinks with diselenides had practically no effect on the 3D structure and inhibitory activity of these Se-Hir analogues, yet still demonstrated the general improvement in folding efficiency observed for all the Se-Hir analogues studied here. These results indicate that although disulfide-to-diselenide substitution proved to be generally advantageous for the preparation of hirudin analogues, as in other proteins, the exact location of the replaced crosslink could be important to fully retain its inhibitory activity.

The choice of hirudin as a model in this study seems to have implications beyond its specific folding mechanism. Unlike BPTI, the defining characteristics of hirudin lies in the complex route it follows to achieve its native fold, and this seemingly chaotic pathway is shared by many other proteins[3–5]. Furthermore, many proteins adopt a folding mechanism that lies somewhere on the spectrum between these two opposite folding models[4]. We found that the substitution of native/non-native disulfide with diselenide bonds typically enhances protein folding and yield, preserves protein structure and often confers no significant effect on function. Based on these results, the current study of the seleno-hirudin analogues extends the previous studies on selenocysteine-containing analogues of BPTI[34,35], conotoxins[28–31] and insulin[53–55], which together demonstrate the high potential of disulfide-to-diselenide substitutions in basic and applied research. In addition to its obvious advantage in the study and preparation of disulfide-rich proteins, such an approach may also prove useful for a wide range of applications, including the rational improvement of the stability, activity and specificity of this unique and medically important family of proteins.

## Methods

### General procedure for Fmoc-SPPS.
Peptides were prepared manually or using an automated peptide synthesizer (CS136XT, CS Bio Inc. CA) typically on 0.25 mmol scale. Fmoc-amino acids (2 mmol) were activated with HCTU (2 mmol) and DIEA (4 mmol) for 5 min and coupled for 25 min, with constant shaking. Fmoc deprotection step was carried out with 20% piperidine in DMF for 2 × 5 min, and DMF was used for washing the resin. Fmoc-Sec(MoB)-OH was coupled manually using DIC/OxymaPure activation method[44].

The cleavage from resin was performed using TFA:water:thioanisole (94:3:3) cocktail for 3–4 h, and in the presence of 2 equiv of DTNP[56] if Fmoc-Sec(MoB)-OH was present.

The conversion to thioester was done by dissolving the peptide in PB buffer (200 mM, 6 M Gn·HCl, pH ~2.5) and treated with 50 equiv of acetylacetone (acac) and 200 equiv of MPAA for 4–6 h at room temperature[43].

### Native chemical ligation (NCL).
Native chemical ligation (NCL)[37,38] was used to ligate pure segments and the preparation of hirudin and its Sec-containing analogues. This reaction was performed in degassed phosphate buffer (100 mM PB, 6 M Gn•HCl, 50-200 mM MPAA, pH~7.3).

### Oxidative folding.
The oxidative folding experiments were performed according to the reported study by Chang et al., to allow for a direct comparison[16]. All folding reactions were performed under anaerobic conditions (except if stated otherwise) in an anaerobic chamber (Coy Laboratories Inc., O₂ sensor kept at <5 ppm) with nitrogen and hydrogen atmosphere (95%:5%) in degassed Tris·HCl buffer (100 mM Tris·HCl, 200 mM NaCl, 1 mM EDTA, pH 8.7). Oxidized glutathione (GSSG, 5 equiv, final concentration 150 µM was added to 30 µM of reduced WT-Hir and its seleno-analogues. At various time intervals, 80 µl aliquots were removed and quenched with 30 µL of 2 M HCl, and stored at −20 °C before analysis by analytical HPLC. The reaction mixture was injected into Atlantis T3 column (3 µm, 4.6 × 150 mm heated to 40 °C) and eluted from the column by 15:85 to 22:78 gradient over 15 min (B:A), and increasing to 28:72 over 32 min, and reaching the initial conditions over 36 min. All chromatograms were monitored at a wavelength of 220 nm (Fig. 3 in the main text).

The same experiment was repeated under aerobic conditions and in the absence of GSSG with WT-Hir and Hir(C16U/C28U) (Figs. S13 and Fig. 4a in the main text), in which oxygen is the only oxidant in solution.

### Peptide and protein characterization.
Peptides were analyzed using reversed-phase high performance liquid chromatography (RP-HPLC) by analytical Waters Alliance HPLC or Acquity UPLC H-Class with UV detection (wavelength 220 nm and 280 nm). Columns: XSelect C18 column (3.5 µm, 130 Å, 4.6 × 150 mm) for peptide analysis and C4 column (3.5 µm, 4.6 × 150 mm) or C8 (3.5 µm, 4.6 × 150 mm) for ligation reactions. Atlantis T3 column (3.5 µm, 2.1 × 100 mm) was used to analyze protein folding. Electrospray ionization mass spectrometry (ESI-MS) was performed on LCQ Fleet Ion Trap MS (Thermo scientific). Peptide masses were calculated from the experimental mass to charge ($m/z$) ratios from all of the observed multiply charged species of a peptide. Deconvolution of the experimental MS data was performed with the help of MagTran v1.03 software.

### Peptide and protein purification.
Purification was performed on preparative or semi-preparative Waters 150Q LC system, using XSelect C18 column (5 µm, 130 Å, 30 × 250 mm), XBridge BEH300 C4 column (5 µm, 19 × 150 mm) and C8 (5 µm, 10 × 150 mm). Linear gradients of acetonitrile (with 0.1% TFA) in water (with 0.1% TFA) used for all systems to elute bound peptides. The flow rates were 1 mL/min (analytical, column heated at 30 °C), 10 mL/min (C4 semi-preparative), 20 mL/min (C18 preparative) and 3.5 mL/min (C8 and C4 semi-preparative).

### X-ray crystallography.
The lyophilized Se-hirudin (Se-Hir) protein analogues were dissolved in water to a final concentration of 10 mg/mL. Thrombin from bovine plasma (Sigma-Aldrich) was dissolved in a solution containing 0.08 M sodium phosphate buffer (pH 7.5), 0.5 M NaCl and 0.1% PEG 6000 to prepare an 8 mg/mL protein solution. The three Se-Hir analogues: Hir(C16U/C28U); Hir(C6U/C14U); Hir(C22U/C39U), prepared in this way were then mixed with the thrombin solution (in molar ratio of 1:1.3 for Thrombin/Se-Hir analogue) to obtain the three Se-Hir-Thrombin complexes submitted for crystallization.

### 2D-NMR of WT-Hir, Hir(C6U/C14U) and Hir(C6U/C16U).
WT-Hir, Hir(C6U/C14U) and Hir(C6U/C16U) samples were all prepared with identical conditions. Proteins were dissolved in 320 µL of 10% D₂O in filtered TDW where the pH was adjusted to 4.5 using solutions of 0.5 M NaOH and 0.1 M HCl. The final ionic strength was approximately 7 mM for all samples. The final concentration of WT-Hir was 1.44 mM, Hir(C6U/C14U) was 1.41 mM and Hir(C6U/C16U) was 0.6 mM.

The experiments were performed under identical conditions on a Bruker AVII 500 MHz spectrometer operating at a proton frequency of 500.13 MHz, using a 5-mm selective probe equipped with a self-shielded XYZ-gradient coil at 18.2 °C. Phase sensitive double quantum filtered correlation spectroscopy (DQF-COSY) experiments[57] were acquired using gradients for water saturation. Spectra were processed, analyzed and presented with TopSpin (Bruker Analytische Messtechnik GmbH) and NMRFAM SPARKY software[58].

### Inhibition assays.
Thrombin activity was assayed in Tris-HCl buffer (50 mM, 154 mM CaCl₂, 0.2% polyethylene glycol 6000, pH 8) at 37 °C[13,50]. Following pre-incubation of 184 pM of thrombin and inhibitor with a concentration varying between 0 and 3.7 nM in a total volume of 0.30 mL. The enzymatic reaction was

started by the addition of 68.5 µM of N-(p-Tosyl)-Gly-Pro-Arg-p-nitroanilide (Tos-Gly-Pro-Arg-NH-Np). The initial rate of p-nitroaniline formation was followed at 405 nm ($\mathcal{E}_{405} = 9920\ cm^{-1}M^{-1}$) using a Thermo Scientific Evolution$^{TM}$ 201 UV–Visible spectrophotometer. Protein concentration was determined by spectrophotometer ($\mathcal{E}_{280}$ of thrombin is 72,150 $cm^{-1}M^{-1}$; $\mathcal{E}_{280}$ of WT-Hir and its seleno-analogues is 2560 $cm^{-1}M^{-1}$). The data were fitted to the following inhibition equation to calculate $K_I$.

$$v = \left(\frac{vo}{2E}\right)\left[\left(\sqrt{(K_I + I - E)^2 + 4K_IE}\right) - (K_I + I - E)\right]$$

## Data availability

The coordinates for the X-ray structures have been deposited to the Protein Data Bank (PDB) with accession PDB IDs: 7A0D for Hir(C16U/C28U)-Thrombin complex, 7A0E for Hir(C6U/C14U)-Thrombin complex and 7A0F for Hir(C22U/C39U)-Thrombin complex.

All other data supporting the findings of this study are available within the paper and its supplementary information files.

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

## Acknowledgements
We thank Mrs. Ricki Notis Dardashti for input on the manuscript. N.M. acknowledges the financial support of Israel Science Foundation (1072/14 and 783/18) and ICRF Acceleration Grant. G.S. acknowledges the support of both the Israel Science Foundation (1905/15) and The Israeli Ministry of Science (3-12484/15). R.M. acknowledges the support of the VATAT scholarship for Arab students. T.H. is grateful to the Neubauer Fellowship for Arab Ph.D. students. We thank Neil Patterson and David Aragao from the Diamond light source (London) for carrying out the data collection for Hir(C6U/C14U)-thrombin (Complex-2) remotely at the Diamond light source (performed within the CCP4 workshop organized at the Ben-Gurion University by Dr. Anat Shahar).

## Author contributions
R.M. and N.M. designed the research. R.M. performed the experiments. L.K. assisted with the experiments. T.H., S.P., S.H. and G.S. performed the crystallography studies and the data analysis. D.E.S. performed the 2D-NMR and analyzed the data. R.M. and N.M. wrote the paper with contribution from all co-authors. All authors reviewed the manuscript.

## Competing interests
The authors declare no competing interests.
