## [Peer Review File · Communications Chemistry]

Reviewers' comments:

Reviewer #1 (Remarks to the Author):

Very nice and in-depth study from the Metanis lab on Hirudin, a 65-residue long peptide/protein with three disulfide bonds and a complex folding pattern. Mousa et al. chemically synthesised WT Hirudin and four seleno-analogues of hirudin by replacing pairs of cysteine residues with selenocysteines. They use a combination of Fmoc-SPPS and NCL to produce these analogues and to systematically study their folding kinetics and patterns by HPLC and MS. Furthermore they were able to get high-resolution crystal structures of the seleno-analogues bound to thrombin and carry out an in-depth structural analysis to support the observed bioactivities and physicochemical properties of the analogues. Key outcomes of this study are that complex protein folding can be accelerated and simplified using strategic placement of selenocysteine residues into the synthetic scheme. This also improved yields significantly. This has been shown with other peptides, but Hirudin is clearly a key folding model peptide/protein that adds further details and confirmation for this approach. Interestingly, two of the analogues seem to have slight structural changes that are observable by HPLC retention times as well as by inhibitory activity. Mousa et al. then went through an extremely detailed structure-activity-relationship analysis to understand and explain these observed differences, which was a pleasure to read and constitutes of a large body of work including multiple X-ray structures and NMR analyses.

Overall, excellent work, and I thus recommend this work for publication, with minor revisions.

A few questions, suggestions and minor revisions:

Questions:

It is not clear to me (at least not following the results section of the synthesis and folding details), on how you can exclude (or have excluded) a potential Se-S, Se-S formation instead of the expected Se-Se, S-S formation with your 6-14 and 6-16 analogues. This explanation is missing in the results section. Considering that Cys14 and Cys16 are very close together, how confident are you with the assignment in this regard? Same for the 6-16, how do you know that this is a SS plus 2xSe-S and not 2x SS and 1x SeSe, particularly considering that the HPLC trace would rather support a misfolding / non-native bond formation?

Do you observe any folding dimers or oligomers induced by the diselenide or selenylsulfide bond (e.g. after cleavage or during oxidation)?

I don't think that oxidized GSSG should be called the oxidant. Even under anaerobic conditions and after buffer purging you will still have oxygen in the aqueous buffer and the driving folding force is the pH. GSH might be released during the folding process, which might speed up the folding and also induce disulfide bond scrambling. Better to call it folding catalyst (which you do later) or scrambling agent.

Is there a reason why GSH is not added to the folding mixture (often added in combination with GSSG)?

What is the surface exposure of the 6,14/16 bonds? Diselenide bonds are slightly more hydrophobic than SS bonds, which could play a role (or at least explain the HPLC retention time shift) if they are surface exposed.

Is there a chance that the SS at 6,14/16 are somehow involved in the binding/inhibition of thrombin?

Suggestions:

Title:

I think Hirudin should be mentioned in the title.

A suggestion:

Enhanced and simplified folding of Hirudin driven by diselenide bond formation.

I would remove the outcome summary at the end of the introduction as this already gives away the results section to come. Instead highlight the research questions and aims of this study leading up to the result section. That will keep the reader hooked.

Figure 1. and general suggestion:

I think it is important to present the complete sequence of Hir either in Figure 1 or Scheme 1.

For better clarity, I would suggest to move away from using the position numbers to indicate the SS / SeSe connectivity. I would simply establish early on what the Hirudin native connectivity is using Cys residues numbering starting from the N-terminus, i.e. C1, C2, C3, etc. C1-2, C3-5, C4-6; or U3-5, etc. This will make it easier to follow the results and discussion.

Figure 1b already presents results (folding pattern of seleno analogues and should be moved to the result section).

It would be nice to highlight the time and yield improvements compared to WT-Hir in a table better, as this is a key message of the paper (maybe even including the one under aerobic conditions without GSSG, which is currently in the SI).

Scheme 1, could be presented nicer and include the reaction conditions / chemicals.

Also the scheme suggests that you get fully reduced seleno-Hir, which is not correct, as the first Se-Se is readily formed. This should be indicated in the scheme. Maybe this scheme/figure can be expanded to actually show the sequence of Hir and present all the produced analogues with their identified SS, SeSe, SeS connectivities. That would improve the clarity and help to follow the folding complexity.

I agree overall with the in-depth discussion in the SI – maybe some of it could be brought back into the main manuscript if space allows (e.g. this below or other important conclusions).

It should be noted, however, that the similarities observed in the current structures are based on the hirudin inhibitors when they are tightly bound to their target protease, thrombin. This is an important point, since the exact structure of the specific hirudin analog examined may, in principle, be different when it is free and unbound, since such protein-inhibitor binding may force the bound hirudin into the specific conformation that is energetically preferred by the tight hirudin-thrombin interactions. Saying that, it should also be mentioned that the an NMR analysis in solution, confirmed that the solution structure of the free Hir(C6U/C14U) analog is practically identical to the corresponding free WT-Hir in the same conditions, generally addressing this potential concern.

Figure 4. Could be expanded and have the disulfide regions zoomed in underneath the three structures to provide better insight in the regions of interest. This would also go well with an expansion of the interesting structural discussion and conclusion that is currently in the SI.

Minor revisions:

which plague its folding rate and yield in vitro – could be expressed/rephrased better.

At the end (instead of ends)

color / analog vs colour / analogue – mix of UK and American spelling, please fix.

What was the rationale for the Sec 6-16 study (and why not other non-native ones)? The 6-16 bond is already introduced at the end of the introduction and leaves the reader puzzled on why this and not another one. Either add already a rationale here or just introduce the non-native one later.

Rephrase the historical writing style: The current study began ...

Add rationale for the ligation site.

Specify what thioester surrogate

It would be informative to mention how many potential folding products exist with three disulfide bonds present to better set the scene for the folding complexity (could be added to the intro or in the folding section).

Figure 3. Increase font size and spacing of legend and K_i values. Fix alignment.

Supporting Information

Synthesis of Hir(29-65) Scheme:

Scheme / Figure name and number missing – this applies also to all the other schemes in SI.

Fmoc-NH-Gln(Trt)-Wang-Resin

Gln, the I is bolded

mmol/g and not /gr

on an automated peptide synthesizer

double coupled

Figure S1:

what HPLC wavelength?

obs? monoisotopic mass? M+H? please be specific.

please address this also for the other figures.

Fmoc-Sec(Mob)-OH – capitalise Mob throughout manuscript.

SeH is indicated to be present, which is however incorrect as it directly forms a diselenide or selenylsulfide bond. Needs to be presented correctly.

Figure S7.

purified with a single diselenide

why do you only have three reduced thiols and not four?

Figure S9

semi-prep

Hir(C22U/C39U) scheme – bond lengths are all over the place, sometimes bold, different lengths, overlapping numbers. In later schemes red colour is used for the Se bond etc. – these schemes need some attention.

Figure S12 has lines from cutting peaks in the figure.

Has the MS been taken from the whole purified fraction? It seems that you have a Glu deletion in there (big shoulder in analytical HPLC), however this impurity does not show up in the mass spec. Final mass specs should be taken and presented from the whole purified compound that was used in the study.

Figure S15 still has some cut and paste lines in the figure that should be removed / cleaned up.

Table S2 should be fitted within the text width. I would remove the two decimal numbers and also highlight any obvious differences if there are any. Figure S19 should also be within the text margins (or text margins could be expanded, since quite broad margins / lots of white space).

Reviewer #2 (Remarks to the Author):

In this paper, Metanis et al have demonstrated the use of diselenide bridges to direct folding of thrombin inhibitor Hirudin, a mini-protein that encompasses three disulfide bonds. The authors synthesised three different Hirudin analogues, where each S-S bond was substituted by a Se-Se bond. The authors showed that the rate of protein folding is improved in all three cases, and that all three diselenide analogues presented similar tridimensional structure as the native protein upon complexation with thrombin. Two out of the three Sec-analogues exhibited similar thrombin inhibitory activity as the parent protein, whereas Sec-replacement at the 6-14 position significantly decreased activity.

Overall, the manuscript is clearly written and demonstrated results that will be of interest to peptide and protein chemists working with disulfide-rich molecules. However, the manuscript lacks novelty for the general chemistry community, as diselenide-directed folding has been already reported for other peptides and proteins by the same group and others (i.e, to hormones, other toxins, insulin, BPTI). The fact that diselenides can improve folding of disulfide-rich peptides, mostly without impeding native structure and activity, has been already established in several other publications over the past years (for instance, refs. 25-34,52-54).

In the case of Hirudin, which naturally folds quite efficiently, the Se-Se substitution offers just a slightly enhancement in folding. Perhaps, the authors could improve the novelty of the paper by showing the pharmacological consequences of the diselenide derivative over the native inhibitor.

Reviewer #3 (Remarks to the Author):

General comments:

Hirudin is a potent thrombin inhibitor with a compact N-terminal domain with three disulfide bonds. The formation of native and non-native disulfide networks during *in vitro* folding traps the protein in different disulfide isoforms leading to poor yields. In this manuscript, Norman Metanis and colleagues have shown that replacing one of the native disulfide by diselenide leads to enhanced rate of folding of hirudin. This strategy has been used in protein folding for more than two decades (Moroder L. Isosteric replacement of sulfur with other chalcogens in peptides and proteins. *J Pept Sci.* 2005;11(4):187-214; Muttenthaler M, Alewood PF. Selenopeptide chemistry. *J Pept Sci.* 2008;14(12): 1223-1239; and several examples (apamin, BPTI and insulin) from Metanis' lab). Although this manuscript provides detailed characterization and evaluation of the structure of the hirudin-thrombin complexes, the strategy/ idea cannot be considered as innovative. The data, however, highlights the advantages and limitations of diselenide chemistry in protein folding.

The relatively stronger diselenide, compared to disulfide, most likely forms the bond first and thus, making the hirudin folding problem from three to two disulfide bond problem. This is reflected in the enhanced rate of folding. The non-native diselenide isoform appears (indicated by distinct elution profile) to remain folded in a non-native fold. Slightly larger size of selenium results in longer (10-20%) diselenide bonds, in some cases, affects the function (native C6U/C14U isoform shows about 20-fold loss). In most proteins disulfides are packed inside the protein core and the larger size, at times, may not allow adequate packing space for diselenide bonds within the core. In such proteins (particularly in smaller disulfide-rich domains) the core packing may severely affect the protein conformation and hence, the function. Relatively higher hydrophobicity of diselenide may affect folding of proteins (although rare) with surface disulfides.

The authors should highlight these and other advantages and limitations of diselenides in folding, form and function of proteins.

Specific comments:

Introduction: The authors indicate that the folding provides "in-depth elucidation of the folding process." It is important to indicate that these studies provide information of in vitro folding and not necessarily that of in vivo folding.

Figure 2: In only C6U/C14U, most folding intermediates 'disappear' by 90 min. However, in other isoforms, these intermediates linger much longer. Why?

Figure 2e: The identify a peak (marked with #) that elutes ahead of folded conformation as that of a contaminant of unknown mass. It may be interesting to identify the structure, at least with MS data.

Figure 2f: Unlike other native diselenide isoforms, C6U/C14U elutes with a different retention time indicating slightly altered conformation of this isoform. The authors should examine these changes in free and bound forms C6U/C14U and compare the structure with other isoforms.

Figure 2f: The non-native diselenide C6U/C16U also elutes with distinct retention time. It will be important to identify the disulfide/diselenide (or mixed) linkages in this isoform. It is unclear whether in this isoform diselenide reshuffles to form native linkages.

Figure 4a and 4c: The diselenide bonds connect residues that are separated by 11- and 17-residue segments. It is not clearly 'visible' in the orientation as shown. Please choose the orientation that allows the reader to see this aspect.

The authors should highlight the advantages and limitations of diselenides in folding, form and function of proteins, as mentioned above.

A point-by-point letter to reviewer's comments.

Reviewer #1

Very nice and in-depth study from the Metanis lab on Hirudin, a 65-residue long peptide/protein with three disulphide bonds and a complex folding pattern. Mousa et al. chemically synthesised WT Hirudin and four seleno-analogues of hirudin by replacing pairs of cysteine residues with selenocysteines. They use a combination of Fmoc-SPPS and NCL to produce these analogues and to systematically study their folding kinetics and patterns by HPLC and MS. Furthermore they were able to get high-resolution crystal structures of the seleno-analogues bound to thrombin and carry out an in-depth structural analysis to support the observed bioactivities and physicochemical properties of the analogues. Key outcomes of this study are that complex protein folding can be accelerated and simplified using strategic placement of selenocysteine residues into the synthetic scheme. This also improved yields significantly. This has been shown with other peptides, but Hirudin is clearly a key folding model peptide/protein that adds further details and confirmation for this approach. Interestingly, two of the analogues seem to have slight structural changes that are observable by HPLC retention times as well as by inhibitory activity. Mousa et al. then went through an extremely detailed structure-activity-relationship analysis to understand and explain these observed differences, which was a pleasure to read and constitutes of a large body of work including multiple X-ray structures and NMR analyses.

Overall, excellent work, and I thus recommend this work for publication, with minor revisions.

Our response: We thank the reviewer for supporting the publication of our manuscript and for the very useful and detailed suggested revisions. We are certain that these revisions have strengthened our manuscript immensely.

Questions and suggested revisions

1. It is not clear to me (at least not following the results section of the synthesis and folding details), on how you can exclude (or have excluded) a potential Se-S, Se-S formation instead of the expected Se-Se, S-S formation with your 6-14 and 6-16 analogues. This explanation is missing in the results section. Considering that Cys14 and Cys16 are very close together, how confident are you with the assignment in this regard? Same for the 6-16, how do you know that this is a SS plus 2xSe-S and not 2x SS and 1x SeSe, particularly considering that the HPLC trace would rather support a misfolding / non-native bond formation

Our response: Selenols are prone to oxidation, and two selenols tends to oxidize rapidly forming a diselenide bridge which is preferred over disulphide bridges and selenylsulfide formation in peptides/ proteins. This is known for many of the proteins tested in our group and other research groups. MS data supports our analysis. In all Se-

hirudin analogues, a single bond was oxidized after purification of the corresponding ligated proteins (-2 Da from the mass of fully reduced protein), while the WT-Hir was isolated as fully reduced. Since the Se-hirudin analogues contain 4 Cys and 2 Sec residues, the only option is to have a single diselenide bond and 4 free thiols. Furthermore, after folding, we observed the mass of the proteins was -6 Da from the fully reduced protein, suggesting the formation of 3 S-S bonds (for WT-Hir) or a Se-Se and 2 S-S bonds (for Hir(C6U/C14U), Hir(C16U/C28U) and Hir(C22U/C39U)). This was unequivocally supported by catalytic activity and crystal structure analysis of the Se-hirudin analogues.

In case of the Hir(C6U/C16U) analogue, the protein was purified after NCL with a single non-native diselenide at position 6-16 (-2 Da from fully reduced protein), similar to all other seleno-hirudin analogues. However, after folding the final native structure was obtained (-6 Da from fully reduced protein), as confirmed by HR-MS indicating the formation of three crosslinks. The 10-fold difference in activity is too small to indicate a formation of a wrong folded state (the Hir(C6U/C14U) had a 20-fold lower activity and still had a perfect native structure). This suggest that native state was formed with this analogue as well, but in this case contained two selenylsulfides at positions Sec6-Cys14, Sec16-Cys28 and a single disulphide at Cys22-Cys39.

It is worth noting that all seleno-hirudins analogues were purified and confirmed by HR-MS (See figure S9-12 in SI).

We explained this extensively in the SI and in main text in the Results and discussion section.

2. Do you observe any folding dimers or oligomers induced by the diselenide or selenylsulfide bond (e.g. after cleavage or during oxidation)?

Our response: A single peptide was shown to form a dimer after cleavage and purification, the Hir(39-65)(C39U), since it does not contain other Cys or Sec residues in its sequence. Fig. S2 show the HPLC and ESI-MS of this peptide after treatment with TCEP and sodium ascorbate, which reduces the diselenide bonds. We added this sentence in the SI (Fig.S2) to clarify this important point.

All other peptide segments containing Sec residues were isolated as monomers, with a single diselenide bonds, and in one case, Hir(1-38)(C22U)-COSR (used in the ligation with Hir(39-65)(C39U) to give the Hir(C22U/C29U) analogue), with a mixture of selenylsulfide containing isomers. This is clearly mentioned in the legend of Fig. S5, in SI, where a broad peak for the Hir(1-38)(C22U)-COSR was observed suggesting the formation of a mixture of selenylsulfide containing isomers.

During folding experiments the formation of diselenide containing dimers/oligomers is tentatively possible, but we never observed that, as supported by MS analyses of all the intermediates formed.

3. I don't think that oxidized GSSG should be called the oxidant. Even under anaerobic conditions and after buffer purging you will still have oxygen in the aqueous buffer and the driving folding force is the pH. GSH might be released during the folding process, which might speed up the folding and also induce disulphide bond scrambling. Better to call it folding catalyst (which you do later) or scrambling agent.

Our response: Most of our folding experiments were performed under anaerobic conditions, in anaerobic chamber (Coy Laboratories Inc., under N₂ and H₂ atmosphere) after degassing of buffers and all solutions (O₂ sensor set at <5 ppm). We are o.k. with calling oxidized glutathione, GSSG, as the oxidant in these folding experiments. A better name for it would be “additives”. We have now included the expression “folding catalyst” in the manuscript as requested. Although a catalyst normally used for components that are not consumed during the reaction, and are used in sub-stoichiometric amounts, which is not the case here.

In one case we compared the folding of WT-Hir and Hir(C16U/C28U) under aerobic conditions, in the absence of GSSG, and with oxygen as the solely oxidant in solution. As it is clearly seen (Fig. S13 and in Table 1 in the main manuscript), oxygen is not a superior oxidant for the folding of hirudin, or Se-hirudin, although still more efficient for the Se-hirudin.

We agree that the released GSH during folding will “catalyze” thiol-disulphide (diselenide) exchange reactions, and shuffling of intermediates, including the scrambled isomers that predominates the late stage folding of hirudin. We have revised the manuscript accordingly to explain this important point.

4. Is there a reason why GSH is not added to the folding mixture (often added in combination with GSSG)?

Our response: We agree with the referee that usually folding buffers contain a mixture of oxidized and reduced additives, such as cysteine/cystine, GSH/GSSG, etc. However, in our case we followed the conditions used by Chang et al (see ref. 15 in the manuscript) for the folding of hirudins, to have little changes to the known procedure and allow a direct comparison with that report. The rationale behind choosing these conditions is now indicated in the revised manuscript and the SI.

5. What is the surface exposure of the 6,14/16 bonds? Diselenide bonds are slightly more hydrophobic than SS bonds, which could play a role (or at least explain the HPLC retention time shift) if they are surface exposed.

Our response: We thank the referee for this important point and suggestion. We agree that diselenide bonds are slightly more hydrophobic than the disulphide bonds. We examined the structures of the native state of WT-Hir and the Se-Hir analogues, and in all cases the disulphide/diselenide bonds are mostly buried in the core domain and are not solvent exposed. In the case of Hir(C6U/C14U) and Hir(C6U/C16U), the HPLC retention time was longer than for the other analogues, indicating that these analogues

are more hydrophobic, at least in the unfolded fully oxidized state, as the pH of the HPLC solvents was ~ 2 . We have added this explanation to the manuscript. In addition, Solvent Accessible Surface Area (SASA) calculations were performed for the Wild-type and the Se-Hirudin thrombin complexes using GETAREA webserver (Reference: *Fraczkiewicz, R. and Braun, W. (1998) "Exact and Efficient Analytical Calculation of the Accessible Surface Areas and Their Gradients for Macromolecules" J. Comp. Chem., 19, 319-333*). The calculations showed that both cysteine (Cys) and selenocysteine (Sec) residues at position 39 are the most exposed compared to positions 6, 14, 16, 22 and 28, of these, the residues at positions 14 and 28 are particularly buried. These calculations are reinforced with the experimental observation of longer HPLC retention time for Hir(C6U/C14U) and Hir(C6U/C16U) analogues. However, there was only a relatively small difference in the ASA between Cys and Sec at the four different positions and when compared to the maximal SASA of cysteine (102.3 \AA^2), given at the GETAREA server site, it appears that all the Cys/Sec residues of hirudin are at least somewhat buried. We added this Figure to the SI (Figure S22).

6. Is there a chance that the SS at 6,14/16 are somehow involved in the binding/inhibition of thrombin

Our response: None of these residues is directly involved in binding of thrombin. Studies have shown that hirudin has high specificity for thrombin, resulting from interaction both at the active site of the enzyme and at the secondary binding site distant from the active site. The residues involved in binding to thrombin are the N-terminus residues: Val1, Val2 and Tyr3 (adjacent to residue 6) which binds to the active site of thrombin, while the C-terminal segment (55-65 AA) binds to the groove on the surface of thrombin.

Other suggestions:

7. Title A suggestion: “Enhanced and simplified folding of Hirudin driven by diselenide bond formation”

Our response: Thanks for this interesting suggestion. We understand that the referee wishes to keep the title more specific for hirudin, as this study selected hirudin as the protein model. However, we are now confident that this strategy can be applied to any disulphide-rich protein, and we explain why. We have shown before that it worked for BPTI, a model folding for many disulphide-rich proteins, and now to hirudin, which is the opposite folding model. This can be also the case for disulphide-rich proteins that have a folding mechanism that is a combination of these two extreme models.

8. I would remove the outcome summary at the end of the introduction as this already gives away the results section to come. Instead highlight the research questions and aims of this study leading up to the result section. That will keep the reader hooked.

Our response: Although this is the case for many journals, *Nature Communication Chemistry* style suggests that the introduction should end with a summary of the key results and conclusions of the work. This comment was supported by the Editor.

Figure 1. and general suggestion.

9. I think it is important to present the complete sequence of Hir either in Figure 1 or Scheme 1.

Our response: We agree with the reviewer, and as such a revised Scheme 1 was added.

10. For better clarity, I would suggest to move away from using the position numbers to indicate the SS / SeSe connectivity. I would simply establish early on what the Hirudin native connectivity is using Cys residues numbering starting from the N-terminus, i.e. C1, C2, C3, etc. C1-2, C3-5, C4-6; or U3-5, etc. This will make it easier to follow the results and discussion.

Our response: There are two ways to present disulphide bonds in disulphide-rich proteins. One by showing the specific position of the Cys residues involved in the disulphide formation, which we adopted in this manuscript. The other option is to show the relative position of Cys residues from the N-terminus, as the reviewer suggested. Both presentations are acceptable. We prefer the first option; however we also now added the disulphide connectivity as the referee suggested in the manuscript to make it clear for the readers who are used to the second option.

10. Figure 1b already presents results (folding pattern of seleno analogues and should be moved to the result section).

Our response: We agree with the reviewer on this point, but we think that this figure also gives a glimpse of the results of this study, as in the case of the outcome summary at the end of the introduction. Additionally, Fig. 1b also show the folding model of WT-Hir (in bold), which we think is necessary to be in the introduction. We think that in the final form of the manuscript, this figure can be moved to be closer to the Results and discussion section. Alternatively, we could change Figure 1b to include only the WT-

Hir folding mechanism, while providing another figure for the Se-Hir analogues in the Results and discussion section if space allowed by the journal.

11. It would be nice to highlight the time and yield improvements compared to WT-Hir in a table better, as this is a key message of the paper (maybe even including the one under aerobic conditions without GSSG, which is currently in the SI)

Our response: We agree with the reviewer and as such revised Table 1 which now includes a column for folding time and two new rows for WT-Hir and Hir(C16U/C28U) in the absence of GSSG.

12. Scheme 1, could be presented nicer and include the reaction conditions / chemicals. Also the scheme suggests that you get fully reduced seleno-Hir, which is not correct, as the first Se-Se is readily formed. This should be indicated in the scheme. Maybe this scheme/figure can be expanded to actually show the sequence of Hir and present all the produced analogues with their identified SS, SeSe, SeS connectivities. That would improve the clarity and help to follow the folding complexity.

Our response: We agree with the reviewer. We revised Scheme 1 in the manuscript accordingly, which now contains the amino acid sequence of WT-Hir, the position of the ligation side, and Cys residues, and a schematic resemble of the seleno-Hir analogues after NCL reaction. The NCL reaction conditions are indicated as well.

13. I agree overall with the in-depth discussion in the SI – maybe some of it could be brought back into the main manuscript if space allows (e.g. this below or other important conclusions) “It should be noted, however, that the similarities observed in the current structures are based on the hirudin inhibitors when they are tightly bound to their target protease, thrombin. This is an important point, since the exact structure of the specific hirudin analogue examined may, in principle, be different when it is free and unbound, since such protein-inhibitor binding may force the bound hirudin into the specific conformation that is energetically preferred by the tight hirudin thrombin interactions. Saying that, it should also mentioned that the an NMR analysis in solution, confirmed that the solution structure of the free Hir(C6U/C14U) analogue is practically identical to the corresponding free WT-Hir in the same conditions, generally addressing this potential concern.”

Our response: We totally agree with the referee. We have moved this important section back to the main manuscript, a section that we previously moved to the SI in order to keep the MS shorter.

14. Figure 4. Could be expanded and have the disulphide regions zoomed in underneath the three structures to provide better insight in the regions of interest. This would also go well with an expansion of the interesting structural discussion and conclusion that is currently in the SI

Our response: We agree with the reviewer, in order to enhance the clarity of the figure based on the comments of both Referee #1 and #3, we have enlarged the individual

panels and added underneath each one a zoomed-in and reoriented image of the diselenide bond region.

Minor revisions:

15. which plague its folding rate and yield in vitro – could be expressed/rephrased better.

Our response: We replaced the word “plague” with “limiting”

16. At the end (instead of ends)

Our response: revised

16. color / analog vs colour / analogue – mix of UK and American spelling, please fix.

Our response: revised. Color, analog, synthesized and disulfides were revised to the UK spelling

17. What was the rationale for the Sec 6-16 study (and why not other non-native ones)? The 6-16 bond is already introduced at the end of the introduction and leaves the reader puzzled on why this and not another one. Either add already a rationale here or just introduce the non-native one later.

Our response: we revised the introduction to explain the rationale behind choosing this specific non-native disulphide bond.

18. Rephrase the historical writing style: The current study began ...

Our response: revised

19. Add rationale for the ligation site.

Our response: the rationale is added.

20. Specify what thioester surrogate

Our response: revised.

21. It would be informative to mention how many potential folding products exist with three disulphide bonds present to better set the scene for the folding complexity (could be added to the intro or in the folding section).

Our response: We agree, and we added a sentence to the introduction section.

22. Figure 3. Increase font size and spacing of legend and Ki values. Fix alignment

Our response: Fig.3 is revised accordingly

Supporting Information

23. Synthesis of Hir(29-65) Scheme:

Scheme / Figure name and number missing – this applies also to all the other schemes in SI.

Our response: revised accordingly.

24. Fmoc-NH-Gln(Trt)-Wang-Resin

Our response: revised.

25. mmol/g and not /gr

Our response: revised.

26. on an automated peptide synthesizer

Our response: revised.

27. double coupled

Our response: revised.

28. Figure S1: what HPLC wavelength? obs? monoisotopic mass? M+H? please be specific. please address this also for the other figures.

Our response: revised, throughout the SI.

29. Fmoc-Sec(Mob)-OH – capitalise Mob throughout manuscript. SeH is indicated to be present, which is however incorrect as it directly forms a diselenide or selenylsulfide bond. Needs to be presented correctly.

Our response: To the best of our knowledge Mob should be capital letter “M” and small letters “ob”, exactly as Trt, Boc, etc. As for SeH, we have revised the scheme accordingly.

30. Figure S7. Purified with a single diselenide why do you only have three reduced thiols and not four?

Our response: Figure S7 is the HPLC and corresponding ESI-MS of the segment Hir(1-38)(C6U/C16U)-Nbz, which has 2 Sec and three Cys residues in its sequence, hence the mass indicate formation of a diselenide and three reduced thiol groups.

31. Figure S9 semi-prep

Our response: revised

32. Hir(C22U/C39U) scheme – bond lengths are all over the place, sometimes bold, different lengths, overlapping numbers. In later schemes red colour is used for the Se bond etc. – these schemes need some attention.

Our response: all schemes in the SI have been revised accordingly.

32. Figure S12 has lines from cutting peaks in the figure. Has the MS been taken from the whole purified fraction? It seems that you have a Glu deletion in there (big shoulder

in analytical HPLC), however this impurity does not show up in the mass spec. Final mass specs should be taken and presented from the whole purified compound that was used in the study.

Our response: We agree with the referee, and Fig. S12 has been revised. We have provided the pure HPLC chromatogram for the compound used in the folding studies. This analogue was prepared multiple times, and sometimes we observed the -Glu side products. For the folding studies we always used the purest fractions isolated after ligation.

33. Figure S15 still has some cut and paste lines in the figure that should be removed / cleaned up.

Our response: Unfortunately, even after zooming on Fig. S15 we could not observe any lines that the reviewer mentioned. Perhaps other Figure?

34. Table S2 should be fitted within the text width. I would remove the two decimal numbers and also highlight any obvious differences if there are any.

Our response: Table 2 is revised accordingly.

35. Figure S19 should also be within the text margins (or text margins could be expanded, since quite broad margins / lots of white space).

Our response: the margins have been revised accordingly.

Reviewer #2

In this paper, Metanis et al have demonstrated the use of diselenide bridges to direct folding of thrombin inhibitor Hirudin, a mini-protein that encompasses three disulphide bonds. The authors synthesised three different Hirudin analogues, where each S-S bond was substituted by a Se-Se bond. The authors showed that the rate of protein folding is improved in all three cases, and that all three diselenide analogues presented similar tridimensional structure as the native protein upon complexation with thrombin. Two out of the three Sec-analogues exhibited similar thrombin inhibitory activity as the parent protein, whereas Sec-replacement at the 6-14 position significantly decreased activity.

Overall, the manuscript is clearly written and demonstrated results that will be of interest to peptide and protein chemists working with disulphide-rich molecules. However, the manuscript lacks novelty for the general chemistry community, as diselenide-directed folding has been already reported for other peptides and proteins by the same group and others (i.e, to hormones, other toxins, insulin, BPTI). The fact that diselenides can improve folding of disulphide-rich peptides, mostly without impeding native structure and activity, has been already established in several other publications over the past years (for instance, refs. 25-34,52-54).

In the case of Hirudin, which naturally folds quite efficiently, the Se-Se substitution offers just a slightly enhancement in folding. Perhaps, the authors could improve the novelty of the paper by showing the pharmacological consequences of the diselenide derivative over the native inhibitor.

Our response: We thank the reviewer for his comments. We agree with the reviewer that this paper should interest the general audience of peptide and protein chemists and chemical biologists in the field of protein folding and beyond. Although diselenides have been introduced and studied in other disulphide-rich proteins, which are all cited in our manuscript, BPTI and hirudin are perhaps the most important proteins, as they are known to be opposite folding models for many disulphide-rich proteins. It is known that many proteins fold in a mechanism similar to BPTI, or hirudin, or a mechanism that is a combination of both. That is why we are confident that this study is really important, as together with our (with Don Hilvert) earlier work on Seleno-BPTI studies, we believe that the use of diselenides for folding can be applied for other proteins, including therapeutic, hard to fold, disulphide-rich proteins. It is true that the folding of hirudin is already efficient, but we chose hirudin not for its yield of folding, but because it is a critical folding model, due to its complex folding behavior.

No further comments were made by the reviewer.

Reviewer #3

General comments:

Hirudin is a potent thrombin inhibitor with a compact N-terminal domain with three disulphide bonds. The formation of native and non-native disulphide networks during in vitro folding traps the protein in different disulphide isoforms leading to poor yields. In this manuscript, Norman Metanis and colleagues have shown that replacing one of the native disulphide by diselenide leads to enhanced rate of folding of hirudin. This strategy has been used in protein folding for more than two decades (Moroder L. Isosteric replacement of sulfur with other chalcogens in peptides and proteins. *J Pept Sci.* 2005;11(4):187-214; Muttenthaler M, Alewood PF. Selenopeptide chemistry. *J Pept Sci.* 2008;14(12): 1223-1239; and several examples (apamin, BPTI and insulin) from Metanis' lab). Although this manuscript provides detailed characterization and evaluation of the structure of the hirudin-thrombin complexes, the strategy/ idea cannot be considered as innovative. The data, however, highlights the advantages and limitations of diselenide chemistry in protein folding. The relatively stronger diselenide, compared to disulphide, most likely forms the bond first and thus, making the hirudin folding problem from three to two disulphide bond problem. This is reflected in the enhanced rate of folding. The non-native diselenide isoform appears (indicated by distinct elution profile) to remain folded in a non-native fold. Slightly larger size of selenium results in longer (10-20%) diselenide bonds, in some cases, affects the function (native C6U/C14U isoform shows about 20-fold loss). In most proteins disulphides are packed inside the protein core and the larger size, at times, may not allow adequate packing space for diselenide bonds within the core. In such proteins (particularly in smaller disulphide-rich domains) the core packing may severely affect

the protein conformation and hence, the function. Relatively higher hydrophobicity of diselenide may affect folding of proteins (although rare) with surface disulphides. The authors should highlight these and other advantages and limitations of diselenides in folding, form and function of proteins.

Our response: We thank the reviewer for these informative comments. As indicated above, in the response to Reviewer #2, and although diselenides substitutions have been introduced and studied in other disulphide-rich proteins (cited in the manuscript), BPTI and hirudin stand out as the **most important and well-studied cases**, representing very different folding models for many disulphide-rich proteins. Although the diselenide bonds are stronger than the original disulphides, this does not make the folding from 3-SS to 2-SS bonds, rather, facilitate the thiol-disulphide exchange reactions, which are central to protein folding. Se-Se bonds form quickly, and can also be reduced in the same manner. The analogue Hir(C6U/C16U), which starts from a non-native diselenide crosslink (6-16), still folds correctly to the native state, containing two selenylsulfide and one disulphide crosslinks.

Regarding the Se atom size and Se-Se bond length, in comparison with S and S-S bond length, we totally agree with the reviewer. It is possible that such changes may affect the structure and as a result the function of the protein. However, all the studies reported before have shown very little changes, either on the structure or the activity. As in the case with the Se-hirudin analogues, our structural studies unequivocally prove that such changes, if found, are relatively minor. The decrease in the activity observed for the Hir(C6U/C14U) and Hir(C6U/C16U) analogs is **only** 10-20 fold lower, which is relatively limited in this range of activity (pM). That is why we did a very deep structure investigation to answer the observed differences in activity with Hir(C6U/C14U). The detailed structural information obtained for the Hir(C6U/C14U) analogue, including the crystal structure of the protein in the bound state with thrombin, and the 2D NMR analysis in the free state in solution, proved that the observed decrease in activity was associated with only relatively minor changes in the 3D structure, and proved that this analogue correctly folded to its native state. These data also suggested that the minor decrease in activity observed for the Hir(C6U/C16U) analogue was probably due to some small structural perturbation around these oxidized bonds, while the protein seemed to fold correctly.

Specific comments:

1. Introduction: The authors indicate that the folding provides "in-depth elucidation of the folding process." It is important to indicate that these studies provide information of in vitro folding and not necessarily that of in vivo folding.

Our response: We have indicated this important point in the introduction. The words "in vitro" were added to emphasize this, and a sentence describing the similarities and differences of *in vivo* and *in vitro* protein folding. A reference for a review by Karan S. Hingorani and Lila M. Gierasch was also added. [ref.#7]

Although it is extremely difficult to compare protein folding *in vivo* with *in vitro*, the physical chemistry behind protein folding still abides. As such it almost impossible to know the exact *in vivo* conditions for protein folding, including salt and protein concentrations, “the crowdedness”, the presence of chaperons, trigger factors, interactions between the protein and the ribosome surface etc.

2. Figure 2: In only C6U/C14U, most folding intermediates 'disappear' by 90 min. However, in other isoforms, these intermediates linger much longer. Why?

Our response: Yes, we can clearly see this in the folding profile of this analog, in addition to a fewer number of intermediates formed during the reaction. The reason for that was described in the manuscript here: “Although the folding yield of the Hir(C6U/C14U) proved to be similar to that of other proteins (Table 1, Fig. 2d, and Fig. S21b), yet, this analogue formed significantly fewer intermediate peaks during the folding process (Fig. 2d). These results are in good correlation with previous kinetic and computational studies, which indicated that not all the disulphide crosslinks have a similar role during the folding process of hirudin. The 6–14 crosslink has been suggested to play a critical role, since it was found to be highly populated among the 1-SS, 2-SS, and 3-SS species in WT-Hir.” This is according to the refs. 16 and 48 in the manuscript. For this reason, most probably that the preformation of the 6-14 diselenide at the early stage direct the rest of the folding process, which simplified the pathway through reduced number of intermediates. This is not necessarily the same for other Se-hirudin analogs, which perhaps go through other, more complex, pathways.

3. Figure 2e: The identify a peak (marked with #) that elutes ahead of folded conformation as that of a contaminant of unknown mass. It may be interesting to identify the structure, at least with MS data.

Our response: Unfortunately, the MS analysis for this peak did not show any detectable mass, in the peptide/protein range. It could have been a small molecule impurity. If it was a protein with specific mass, we would have examined it deeply to test if it was some kind of oxidized species. We are attaching the MS analysis obtained for this peak:

4. Figure 2f: Unlike other native diselenide isoforms, C6U/C14U elutes with a different retention time indicating slightly altered conformation of this isoform. The authors should examine these changes in free and bound forms C6U/C14U and compare the structure with other isoforms.

Our response: We agree with the reviewer. This was the most important analogue that we observed a change in the retention time of the folded state, which raised our concern whether the analogue reached the native state. That is why we followed a deep structural investigation for the bound (by crystallography) and unbound (2D NMR) states of the folded Hir(C6U/C14U). Both crystal structure of the Hir(C6U/C14U)-thrombin complex as well as the 2D NMR of Hir(C6U/C14U) in solution show a very slight changes when compared to the WT-Hir. We have clearly described this in the main manuscript. The 2D NMR spectrum (Fig. S19) is found in the SI.

5. Figure 2f: The non-native diselenide C6U/C16U also elutes with distinct retention time. It will be important to identify the disulphide/diselenide (or mixed) linkages in this isoform. It is unclear whether in this isoform diselenide reshuffles to form native linkages.

Our response: After we have confirmed that the differences in the HPLC retention time observed with Hir(C6U/C14U) analogue **was not for the reason of wrong crosslinks formation** (see above: activity & structure by X-ray and NMR), we were confident that this protein reached the native folded state with the correct disulphide/diselenide crosslinks.

Similarly, the Hir(C6U/C16U) HPLC elution time was difference from WT-Hir (in fact it was between that of WT-Hir and Hir(C6U/C14U)), but very similar activity (only 10-fold higher K_i), also suggest that this protein correctly folded. Please note that crystal structure and NMR studies require very large quantities of the proteins, and at this stage we cannot do it for this analogue (Israel is in lockdown due to COVID19 pandemic). We tried very extensively to do protein digest for many of the Se-hirudin analogs, in order to also prove the formation of the correct crosslinks (at the early stage of the project before we turned to crystallography and NMR studies), but we found that

hirudin is highly stable for digestion, with all the protease used (it is a serine protease inhibitor after all).

For the concern regarding the retention time, please see comment#5 for reviewer #1. Shortly: HPLC solvent are at very acid (pH~2), hence the retention time reflects the differences in the hydrophobicity of the analogues in the **unfolded state (yet fully oxidized)**, which could be different due to the Se presence at different positions.

It is worth noting that differences in HPLC elution time was observed due to diselenide substitutions into proteins for example the Se-conotoxins reported by Alewood (see ref 29). This is also mentioned in the manuscript. We have added a short explanation to the manuscript to emphasize the difference in the elution time observed for Hir(C6U/C14U) and Hir(C6U/C16U).

6. Figure 4a and 4c: The diselenide bonds connect residues that are separated by 11- and 17-residue segments. It is not clearly 'visible' in the orientation as shown. Please choose the orientation that allows the reader to see this aspect.

Our response: We agree with the reviewer. We have taken this opportunity to enlarge the panels of the overall structures in the figure and provided a zoom-in into the orientation of the diselenide bond and the surrounding area. These modifications were carried out taking into account the comment of reviewer #1 (Point 14- Reviewer #1).

7. The authors should highlight the advantages and limitations of diselenides in folding, form and function of proteins, as mentioned above.

Our response: Clearly good points to emphasize. We have clearly indicated these important points in the revised conclusion section of the manuscript.

Reviewers' comments:

Reviewer #1 (Remarks to the Author):

Mousa et al. have addressed all of my comments and I now recommend it for publication. Very good work and a nice read.

I picked up a few minor things that should be addressed in the proof section (or before):

Scheme 1 still indicates the presence of HSe in Hir(40-65), which is not correct. I would keep the X and then indicate either HS or $-2(-Se-$, as done in the SI

I would recommend to use disulfide instead of disulphide – the latter is an older form that is barely used anymore – the majority of work in the field uses disulfide, independent of UK or US English.

Typo: through the release of GSH – instead of thought the release of GSH

SI:

maybe remove the _ in _WT-Hir etc. in Table of Contents

In mass spectra you usually observe the mono-isotopic mass not the average mass as indicated in the Figure captions. Please double check and revise if correct.

Yes I meant MoB (instead of mob – Figure Scheme S4 still has mob)

Reference 11? in 4.4.2.

Reviewer #3 (Remarks to the Author):

The authors have made excellent efforts to answer all questions and comments raised by the reviewers. The revised manuscript is substantially improved.

Specific comments:

I could not find specific structural data to define the disulfide, diselenide and mixed selenylsulfide, particularly in Hir(C6U/C16U) analogue. Structural data (X-ray and NMR) does not cover Hir(C6U/C16U). The authors suggest that this analogue indeed goes from a non-native diselenide (6-16) to correctly folded native state, containing two selenylsulfide (6-14 and 16-28) and one disulphide (22-39) crosslinks. This 10- to 20-fold loss in inhibitory activity could be due to 6-16, 14-28 and 22-39 crosslinks with a minor variation in 6-14 and 16-28 crosslinks. Such crosslinks can be identified using limited reduction of protein using TCEP followed by cyanylation. Cleavage of monocyanylated derivatives by NH_4OH and the MS of the resultant peptides could resolve the crosslinks.

A point-by-point letter to reviewer's comments.

Reviewer #1

Mousa et al. have addressed all of my comments and I now recommend it for publication. Very good work and a nice read.

Our response: We thank the reviewer for supporting the publication of our manuscript and for the very useful and detailed suggested revisions. We are certain that these revisions have strengthened our manuscript immensely.

I picked up a few minor things that should be addressed in the proof section (or before):

1) Scheme 1 still indicates the presence of HSe in Hir(40-65), which is not correct. I would keep the X and then indicate either HS or $-2(-Se-$, as done in the SI

Our response: we have revised Scheme 1 accordingly.

2) I would recommend to use disulfide instead of disulphide – the latter is an older form that is barely used anymore – the majority of work in the field uses disulfide, independent of UK or US English.

Our response: we have replaced all disulphide to disulfide throughout the entire manuscript.

3) Typo: through the release of GSH – instead of thought the release of GSH

Our response: revised

SI:

4) maybe remove the _ in _WT-Hir etc. in Table of Contents

Our response: revised

5) In mass spectra you usually observe the mono-isotopic mass not the average mass as indicated in the Figure captions. Please double check and revise if correct.

Our response: We rechecked and it is actually the average mass not the mono-isotopic

6) Yes I meant MoB (instead of mob – Figure Scheme S4 still has mob)

Our response: revised

7) Reference 11? in 4.4.2.

Our response: revised

Reviewer #2

Reviewer #2 did not send any comments.

Reviewer #3

The authors have made excellent efforts to answer all questions and comments raised by the reviewers. The revised manuscript is substantially improved.

Our response: We thank the reviewer for supporting the publication of our manuscript

Specific comments:

I could not find specific structural data to define the disulfide, diselenide and mixed selenylsulfide, particularly in Hir(C6U/C16U) analogue. Structural data (X-ray and NMR) does not cover Hir(C6U/C16U). The authors suggest that this analogue indeed goes from a non-native diselenide (6-16) to correctly folded native state, containing two selenylsulfide (6-14 and 16-28) and one disulphide (22-39) crosslinks. This 10- to 20-fold loss in inhibitory activity could be due to 6-16, 14-28 and 22-39 crosslinks with a minor variation in 6-14 and 16-28 crosslinks. Such crosslinks can be identified using limited reduction of protein using TCEP followed by cyanylation. Cleavage of monocyanylated derivatives by NH₄OH and the MS of the resultant peptides could resolve the crosslinks.

Our response: We are well aware of the method by Wu and Watson (*Protein Science* 1997) for mapping the disulfide crosslinks using the cyanylation reagent CDAP. I (NM) tried to dissect the crosslinks in BPTI analogs using this method, with no success:

1. Metanis, N.; and Hilvert D. “Strategic Use of Nonnative Diselenide Bridges to Steer Oxidative Protein Folding” (2012) *Angew. Chem. Int. Ed.*, 51, 5585–5588.
2. Metanis, N.; and Hilvert D. “Harnessing selenocysteine reactivity for oxidative protein folding” (2015) *Chem. Sci.*, 6, 322–325.

While it works great for disulfide bonds, it does not work when selenocysteine is present, due to the deselenization reaction of selenocysteine residues into alanine (and serine), please check these references:

1. Metanis, N.; Keinan, E. and Dawson, P.E. “Traceless Ligation of Cysteine Peptides using Selective Deselenization”, (2010) *Angew. Chem. Int. Ed.*, 49, 7049–7053.
2. Dery, S.; Reddy, P. S.; Dery, L.; Mousa, R.; Dardashti Notis, R. and Metanis, N. “Insights into the Deselenization of Selenocysteine into Alanine and Serine” (2015) *Chem. Sci.*, 6, 6207–6212.
3. Reddy, P. S.; Dery, S. and Metanis, N. “Chemical Synthesis of Proteins with Non-Strategically Placed Cysteines Using Selenazolidine and Selective Deselenization”, (2016) *Angew. Chem. Int. Ed.*, 55, 992–995.
4. Dery, L.; Reddy, P. S.; Mousa, R.; Dery, S.; Katorza, O.; Talhami, A. and Metanis, N. “Accessing Human Selenoproteins through Chemical Protein Synthesis” (2017) *Chem. Sci.*, 8, 1922-1926.
5. Mousa, R.; Dardashti Notis, R.; and Metanis, N. “Selenium and Selenocysteine in Protein Chemistry.” (2017) *Angew. Chem. Int. Ed.*, 56, 15818–15827.
6. Dardashti, R. N.; Kumar, S.; Sternisha, S. M.; Reddy, P. S.; Miller, B. G.; Metanis, N. “Selenolysine: A New Tool for Traceless Isopeptide Bond Formation” *Chem. Eur. J.* (2020), DOI: 10.1002/chem.202000310.

To satisfy the concern of Reviewer 3 we decided to prepare more amounts of the analogue Hir(C6U/C16U) and carry on 2D-¹H-NMR COSY analysis of the protein. As it can be clearly seen, this analogue as well as Hir(C6U/C14U) are folded correctly and have similar crosslinks as the WT-Hir, supporting our hypothesis on the correct folding of Hir(C6U/C16U). The effect of the Cys to Sec substitutions in Hir(C6U/C16U) is most obvious in the N-terminal region as was the case for Hir(C6U/C14U). It is also clearly seen that Sec16 has shifted towards high-field. Cys/Sec6 were never observed in any NMR analysis as previously reported Ref. 22. The rest of Cys/Sec residues were clearly assigned. The revised Figure S19 summarizing the results of the three 2D-NMR studies of Wt-Hir, Hir(C6U/C14U) and Hir(C6U/C16U) have been added to the SI.

REVIEWERS' COMMENTS:

Reviewer #3 (Remarks to the Author):

I am fully satisfied the answers and the additional NMR data to support.

I do not have any additional comments.